# Evidence of a Set of Core-Function Genes in 16 Bacillus Podoviral Genomes with Considerable Genomic Diversity

**DOI:** 10.3390/v15020276

**Published:** 2023-01-18

**Authors:** Ahmed Ismail, Tanuj Saini, Ahmed Al Qaffas, Ivan Erill, Steven M. Caruso, Louise Temple, Allison A. Johnson

**Affiliations:** 1Center for Biological Data Science, Virginia Commonwealth University, Richmond, VA 23284, USA; 2Department of Biological Sciences, University of Maryland Baltimore County, Baltimore, MD 21250, USA; 3Integrated Science and Technology, James Madison University, Harrisonburg, VA 22807, USA

**Keywords:** bacteriophage, phage, bacillus, genomics, evolution, podoviruses

## Abstract

Bacteriophage genomes represent an enormous level of genetic diversity and provide considerable potential to acquire new insights about viral genome evolution. In this study, the genome sequences of sixteen *Bacillus*-infecting bacteriophages were explored through comparative genomics approaches to reveal shared and unique characteristics. These bacteriophages are in the *Salasmaviridae* family with small (18,548–27,206 bp) double-stranded DNA genomes encoding 25–46 predicted open reading frames. We observe extensive nucleotide and amino acid sequence divergence among a set of core-function genes that present clear synteny. We identify two examples of sequence directed recombination within essential genes, as well as explore the expansion of gene content in these genomes through the introduction of novel open reading frames. Together, these findings highlight the complex evolutionary relationships of phage genomes that include old, common origins as well as new components introduced through mosaicism.

## 1. Introduction

Bacteriophages (or phages) are viruses that infect a bacterial host. With a worldwide estimate of 10^31^ bacteriophages [1], phages are ubiquitous and are the most abundant genetic reservoir on our planet. Through research-based laboratory courses across the country, the SEA-PHAGES program [2] has facilitated undergraduate researchers making numerous contributions to what we know about genetic diversity of phages that infect a particular host, especially actinobacteriophages [3,4]. Yet, bacteriophage genomes are undersampled compared to their host bacteria, with sequenced bacteria genomes greatly outnumbering phage genomes in GenBank despite the fact that phage genomes are relatively small and easily isolated. Continued discovery of novel phage genomes will illuminate knowledge from this large pool of unexplored genetic diversity [5].

Comparative genomic methods exploring nucleotide and protein sequence relatedness have been established to organize bacteriophage genomes into clusters [6]. Comparative genomics of phage genomes obtained from a particular host system can yield potential knowledge about the dynamics and evolution of the large bacteriophage gene pool. In the largest single host study, 627 Mycobacteriophages were analyzed to reveal that genomes could be clustered but also retain observable genes and groups of genes between different clusters [7]. Relevant to the host system described in this article, Grose et al. [8] investigated the whole genome sequences and proteomes of 93 *Bacillus*-infecting phages, to provide a large-scale framework for understanding the diversity of *Bacillus* phages from diverse host species, including families within the classes *Tectiliviricetes* and *Caudoviricetes*, and including eight *Salasmaviridae* family phages that were in GenBank at the time of publication. Other studies have applied comparative genomics towards large collections of *Enterobacteriaceae* phages [8], *Staphylococcus* phages [9], and *Pseudomonas* phages [10]. These studies have revealed phage populations are dynamic, contain extensive variation in genome properties, and acquire genes through mosaicism from other viruses as well as their bacterial hosts.

The genus *Bacillus* includes many species of Gram-positive, aerobic, endospore-forming bacteria that are ubiquitous in the environment, some of which are human pathogens. Phage phi29 was discovered from garden soil using *Bacillus subtilis* as a host [11], and has become one of the most well-studied bacteriophages in the literature. Most of the protein-coding genes from phi29 have been genetically and biochemically characterized, resulting in detailed understanding of the molecular mechanisms of DNA replication, regulation of transcription, and phage morphogenesis, as well as the development of enzyme-based molecular biology tools. Phages closely related to phi29 were termed the “phi29-like” phages until a recent taxonomic update by the International Committee on Taxonomy of Viruses [12] created the *Salasmaviridae* family. 

We engaged undergraduate students in phage hunting using *Bacillus* species as host bacteria for several years, due to the ubiquitous nature of *Bacillus* in the environment as well as the ease of propagation on simple media. Here, we describe the genome sequences of 16 *Bacillus*-infecting podoviruses obtained from soil samples collected across the mid-Atlantic region using three different *Bacillus* hosts: 12 that were isolated using *B. thuringiensis* as a host, 2 phages from *B. pumilus* and 2 phages from an environmental isolate *Bacillus* sp. 203. These bacteriophages, like phi29, have double-stranded DNA genomes representing two different subfamilies and five different genera within the family *Salasmaviridae*. These phages exhibit podoviral morphology as well as lytic growth properties. Our work offers an in depth look and new insights into the dynamics of genomes of 16 additional *Bacillus*-infecting podoviruses. We explore the comparative genomics of these phages and share observations about similarities and differences within this collection of genomes. While little similarity is observed in whole genome sequence comparisons between groups, a core set of protein functions and a consistent genome organization are maintained in most of these genomes. The genomes we analyze represent an expansion of what is known about the diversity of genes and genomes of phages infecting *Bacillus* hosts.

## 2. Materials and Methods

### 2.1. Host Bacteria

Three different host bacteria were used in phage hunting (Table 1). Twelve of our phages were isolated using *B. thuringiensis* subsp. *kurstaki* ATCC 33679, two were discovered using *B. pumilus* SAFR 32, and one was discovered using an environmental isolate *Bacillus* sp. 203. *Bacillus* sp. 203 was identified as most similar to *B. subtilis* and *B. valezensis* through rDNA sequencing of the host bacteria.

### 2.2. Phage Discovery 

Soil samples were collected by students enrolled in SEA-PHAGES courses at JMU, UMBC and VCU. Phages were discovered from these soil samples through enrichment of extracts with host bacteria and media. Enrichment cultures were filtered with 0.22 μm filters and spotted on a lawn of the appropriate host bacteria. Phage populations were purified by standard SEA-PHAGES methods [13] through at least three rounds of plaque purification by picking a well-isolated plaque into phage buffer, serially diluting, infecting into culture aliquots and plating with top agar. After purification, a plaque was picked into phage buffer and the sample diluted to a concentration predicted to produce near confluent lysis after infection of the appropriate host. Phage lysates were harvested by flooding plates with phage buffer and collecting and filtering (0.22 μm) the lysate. Phage genomic DNA was purified from high-titer lysate using the Promega Wizard DNA purification kit (Madison, WI, USA). Phage particles were applied to formvar/carbon supported copper grids, stained with 0.1% uranyl acetate and visualized through transmission electron microscopy at approximately 60,000× magnification.

### 2.3. DNA Sequencing

Phage genomic DNA was sequenced by Illumina technology (MiSeq or HiSeq 2000) through core facilities located at the University of Pittsburgh Bacteriophage Institute (Pittsburgh, VA, USA), Virginia Commonwealth University Nucleic Acids Research Facility (Richmond, VA, USA), or the North Carolina State Genomic Sciences Laboratory (Raleigh, NC, USA). Reads (150 or 300 bp depending on platform and kit) were assembled into whole genome consensus sequences using Newbler v2.9 or SPAdes [14] v3.12 at ~50× coverage for each assembly. Each genome assembled into a linear sequence, most with defined ends. Genomes were manually examined for coverage and integrity of sequence using Consed v.14 [15], consistent with SEA-PHAGES standards for genome completeness [16]. Short inverted repeat ends were determined by direct inspection of genome sequences. 

### 2.4. Genome Annotation

Genome annotation was completed using DNA Master [17] or PECAAN [18], which integrate gene calling by Glimmer [19] and GeneMarkS [20] for protein-coding genes, and tRNAscan [21] and Aragorn [22] for tRNA genes. Computational predictions were manually curated by students in courses at our three universities consistent with SEA-PHAGES guidelines for annotation [17]. Functions were predicted for genes based on matches to the NCBI Conserved Domains Database [23] with an e-value < 10^−5^; an HHPRED [24] hit with >90% probability, an e-value of <10^−7^, and a peer reviewed publication linked to the PDB hit supporting the function; or the identification of a transmembrane domain by TMHMM [25] and TOPCONS [26]. The pRNA sequences that aid in phage genome packaging [27] were predicted by detecting regions of homology to phi29 and GA-1 [28,29] using BLASTN, along with the presence of promoter [29] and terminator [30] regions, observation of conserved residues, and examination of synteny relationships. RNAFold was used to predict potential pRNA tertiary structures [31]. 

### 2.5. Comparative Genomics

Sixteen genome sequences from phages isolated by us are listed in Table 1 with GenBank identifiers. These sequences were compared to five best BLASTN matches (Baseball_field (MT777452.1), MG-B1 (KC685370.1), GA-1 (NC_002649.1), PumA2 (MN524845.1) and phi29 (EU771092.1)) in dotplot analysis, as well as 5 additional select species to represent each genus in the family *Salasmaviridae* (DK1 (MK284526.1), DLc1 (MW012634.1), Cp1 (AH001309.2), VMY22 (KT780304.1) and B103 (X99260.1)) in ANI and Splitstree analysis.

Whole, concatenated genome sequences were visually compared by dotplot analysis using Gepard (V1.4) [32]. Average nucleotide identity (ANI) values were determined with DNAMaster. Genome maps were examined through Phamerator and a custom *Bacillus* phage genome database [33]. To compare protein content, a spreadsheet listing presence and absence of protein families for each phage was exported from our custom Phamerator database. The file was converted to Nexus format and imported into SplitsTree4 [34] for visualization of protein content as a measure of genome similarity. Evolutionary analysis of protein families was performed using Mega X and inferred by using the maximum likelihood method and JTT matrix-based model [35]. Initial tree(s) for the heuristic search were obtained automatically by applying neighbor join and BioNJ algorithms to a matrix of pairwise distances estimated using the JTT model, and then selecting the topology with superior log likelihood value. Bootstrap iterations were set to 500. Recombination events were visually identified using Phamerator to examine genomes for changes (protein length, genome alignment), and further analyzed using sequence alignment through BLASTN and ClustalOmega [36].

## 3. Results

### 3.1. Phage Isolation and Genome Sequencing

Undergraduate researchers enrolled in a SEA-PHAGES course at three universities isolated and characterized 16 novel phages. A combined 213 undergraduate researchers contributed to the submission of these sequence annotations to GenBank (Table 1). The phages described here were isolated using *B. thuringiensis kurstaki* (12 phages), *B. pumilus* SAFR 32 (2 phages) and an environmental isolate *Bacillus* sp. 203 (2 phages; this host’s closest 16s rDNA matches are *B. subtilis* and *B. valezensis*) as the host bacteria for phage hunting. This group of phages are phi29-like *Caudoviricetes* with podoviral particle morphology (Figure 1) and double-stranded DNA genomes. They are classified as lytic phages due to the apparent absence of genes required for integration as well as consistent clear plaque morphology. Sequenced genomes range in length from 18,642 to 27,206 bp with short inverted terminal repeats, and encode 24–46 genes. GC content ranges from 30 to 39%. Inverted terminal repeats (ITR) for 12 of these genomes were identified through direct inspection of genome ends in assemblies. Two additional genomes revealed ITR sequences that were consistent with homologs at one end, but not on both ends. Each of the ITR sequences that were resolved contain a terminal 5′-TT-3′; presumably the unresolved terminal repeats are due to covalently bound terminal protein blocking full sequencing of end residues [37]. Comparison of these genomes as well as identification of distinct genome features will be described below.

### 3.2. Genome Relationships to Sequenced Bacteriophages

BLASTN was used to determine the suitable related genomes from GenBank for the newly sequenced phages to use in comparative whole genome and protein analyses throughout our work in order to better understand the relationships between these genomes and previously sequenced phages. These genome sequences were incorporated into a dotplot analysis for visualization of sequence identity (Figure 2), and blue boxes were used to highlight likely genus groups within the *Salasmaviridae* family. Ten genomes within the *Claudivirus* genus display high identity with each other (Ademby, RadRaab, StevenHerd11, Stitch, Claudi, KonjoTrouble, Thornton, SerPounce, Aurora and Juan) and exhibit a high query coverage with their best GenBank-related genome Baseball_field (MT777452.1, [38]). The *Claudivirus* genus maintains a high degree of genome identity across their genomes as indicated by the nearly solid diagonal lines in the upper left area of the dotplot. All of these phages were discovered using *B. thuringiensis kurstaki* as host bacteria. BeachBum and Harambe have a strong dotplot relationship, but only 4% query coverage to their best GenBank related genome MG-B1 (KC685370.1). BeachBum and Harambe were discovered using *B. thuringiensis kurstaki* as a host [39], while MG-B1 was discovered using *B. weihenstephanensis* as a host [40]. These genomes are assigned to genera within separate subfamilies in *Salasmaviridae*. Karezi, discovered using a *B. pumilus* SAFR32 isolate as a host, presents a 7% query coverage that is supported by a weak dotplot relationship to phage GA-1 (X96987.2), isolated using *B. subtilis* [41]. Karezi and GA-1 are assigned to the genera *Karezivirus* and *Gaunavirus*, respectively, within the subfamily *Tatarstanvirinae*. Another phage discovered with the *B. pumilus* SAFR32 isolate, WhyPhy, is highly similar to PumA2 (MN524845.1, also discovered using *B. pumilus* [42]) with 95% query coverage, and as visualized by dotplot analysis. These phages have been grouped together into the genus *Bundooravirus*. Whiting18 and Arbo1 sequences are highly similar to the well-studied phage phi29 (EU771092.1), with a query coverage of 97% and extensive dotplot visualization of identity. These genomes are grouped together in the *Salasvirus* genus, and they exhibit no significant nucleotide sequence similarity with other genomes in our study. Whiting18 and Arbo1 were discovered using an environmental isolate *Bacillus* sp. 203 as a host while the original host of phi29 is *B. subtilis* [11]. Despite these sometimes weak relationships between genome sequences from our work and related genomes, comparisons within this group of phages yields valuable observations about other characteristics such as genome structure, synteny, and protein content.

### 3.3. Average Nucleotide Identity

Average nucleotide identity (ANI) was used as a measure of genome-wide similarity (see labels in Figure 2, Appendix A), and this analysis was expanded to include a member of each genus within the *Salasmaviridae* family. The *Claudivirus* genus was organized by ANI values from 87–99%. As the best GenBank match to this group, Baseball_field exhibits an ANI value of 89–96% to genomes in this group, with the highest ANI values towards Claudi, KonjoTrouble, Thornton, and SerPounce. BeachBum and Harambe display ANI values of 96% to each other and exhibit an ANI value of 59% towards MG-B1, suggesting these phage genomes are not closely related to MG-B1. Whiting18 and Arbo1 display ANI values of 99% to each other and 90% to phi29. Karezi and GA-1 share an ANI of 64%, in line with the weak homology observed by dotplot for these two genomes. Phage PumA2 is the best GenBank related genome of WhyPhy with an ANI identity of 86%, reflecting the high level of identity observed between the two genomes by dotplot. Combined, the ANI values support visual observations and groupings of genome similarity by dotplot. ANI values < 60% are observed for phages with little or no sequence identity, as has been previously observed [7].

### 3.4. Protein Content

Protein content analysis provides a third genome-wide measure of similarity. The genomes in this study contain 25–46 open reading frames (Table 1). All the predicted proteins were assigned a protein family (pham) based on a protein sequence having 32.5% identity to at least one other protein sequence, in a custom *Bacillus* phage database in Phamerator [33]. Additional genomes were added to our analysis to represent one member of each genus within the family *Salasmaviridae*: Cp1 (Z47794.1, [43]), DLc1 (MW012634.1, [44]), VMY22 (KT780304.1, [45]), B103 (X99260.1, [46]), and DK1 (MK284526.1, [47]). The presence and absence of 309 phams inferred from 877 total proteins was used as a profile of gene content for each phage, and a splits graph network was displayed using Splitstree [34] (Figure 3). The *Claudivirus* genus displays extensive shared protein content, with twenty-nine protein phams present in all 10 genomes of this group, representing up to 85% of the proteins in these genomes depending on proteome size. MG-B1 may be a subcluster of this group, as indicated by its isolated position on the network. BeachBum and Harambe; WhyPhy and PumA2; and Whiting18, Arbo1, and phi29 contain nearly identical proteomes within those groupings. While Karezi lacked significant similarity to GA-1 by BLASTN, dotplot and ANI analysis, these genomes contain a set of 13 phams that are present in other phages in this collection. Karezi contains a large set of 17 unique phams that make Karezi genome unique. Protein content analysis and Phamerator map comparisons (data not shown) show *Bacillus* phages DK1, DLc1 and VMY22 share limited genome similarity, yet extensive protein conservation and synteny with the *Salasmaviridae* family. Together, protein content analysis supports the groupings previously observed by dotplot and ANI. Finally, it appears *Streptococcus* phages Cp1 and Cp7 were organized into *Salasmaviriade* based on morphological similarity in 1995 [48], and would be better positioned as orphan phage.

### 3.5. Phamerator Maps

Inspection of pairwise Phamerator maps was used as a fourth measure of genome relatedness to confirm relationships identified through dotplot, ANI and protein content analysis (Figure 4 and Appendix A). Overall, this group of genomes maintains a high level of protein synteny and function, especially in the central structural gene region. Other genome regions contain genes that are novel in our database and provide an increase in protein function and sequence diversity to these genomes. The *Claudivirus* genus shares a high level of genome sequence similarity as well as protein content, and a region where the direction of transcription switches in the middle of the genomes that shows expansion of diversity of proteins by the presence of novel genes. 

BeachBum and Harambe show a unique genome organization compared to other phages in this study. They maintain extensive genome sequence and protein content conservation to each other, as described, and ~50% of their protein phams are conserved within only these two phages. Little sequence similarity is observed when compared to MG-B1, though genome map comparison reveals synteny of a significant set of proteins (Appendix A). BeachBum and Harambe genes are primarily transcribed from one of the strands of the genome, a deviation from the transcriptional organization of other phage genomes in this study. 

WhyPhy and PumA2 maintain extensive conservation of genome sequence and protein content, with minor regions of sequence difference and the insertion of one protein in PumA2 that is not present in WhyPhy. Similarly, Whiting18, Arbo1, and phi29 genome sequences and protein content are highly conserved, with only minor variations in genome sequence observed. In contrast to these pairs, Karezi exhibits low sequence conservation to GA-1, but moderate conservation of protein content, and a high degree of synteny and function with GA-1 and WhyPhy. WhyPhy, Whiting18, Arbo1, and Karezi exhibit transcriptional organization similar to the *Claudivirus* genus, in contrast to BeachBum and Harambe. 

Importantly, manual inspection of this set of 16 genomes (Appendix A) reveals each contains a core set of 15 protein functions (Figure 4) and maintains a remarkable conservation of synteny and function despite low nucleotide identity relationships. Our dataset contains a total of 569 proteins, sorted into 162 phams, including 71 orphams. Function predictions obtained through bioinformatics tools reveal potential functions for 40–75% of the proteins depending on the genome. Five protein phams are conserved in all 16 phages in our analysis, as well as other *Salasmaviridae* phages. These protein phams encode DNA polymerase (phi29 gp 2), major head protein (phi29 gp 8), tail knob protein (phi29 gp 9), upper collar protein (phi29 gp 10), and DNA encapsidation ATPase (phi29 gp 16). These proteins have been functionally characterized in phi29: the DNA polymerase catalyzes initiation and elongation of replication for the double-stranded DNA genomes [49]; major capsid protein serves as a structural protein [50]; upper collar protein binds pRNA [51]; and tail knob and DNA encapsidation ATPase control movement of the DNA into the host bacteria. Further examination reveals multiple protein phams encode 10 other proteins present in each genome: structural proteins (phi29 gp 7 scaffold protein and phi29 gp 11 lower collar protein), dsDNA binding protein (phi29 gp 6), terminal protein (phi29 gp 3), two transcriptional regulators (phi29 gp 4 late activator protein and phi29 gp 16.7 early protein), pre-neck appendage protein (phi29 gp 12) involved in host recognition, morphogenesis protein (phi29 gp 13), and the lysis proteins endolysin (phi29 gp 15) and holin (phi29 gp 14). While multiple protein phams appear to be providing the same function, protein sequences can be quite different between phams. For example, the morphogenesis protein cleaves the bacterial cell wall to allow phage DNA injection during infection [52], and in these phages is encoded by two different protein phams with a range of pairwise amino acid identity that can be as low as 26% between phams. Despite the low sequence identity, each morphogenesis protein is predicted to encode M23 metallopeptidase (cd12797) and lysozyme-like (pfam19013) domains. In a second example, terminal protein recognizes a 3′-terminal pair of T residues. Three phams encoding terminal proteins were identified in our phages, and while their sequences range from 15 to 100% amino acid identity, all are predicted to encode a terminal protein domain pfam05435 similar to phi29 gp 3 [53]. Together, it is presumed this set of fifteen conserved protein functions, with sometimes low levels of amino acid sequence identity, represents the core functionality of these phages.

### 3.6. Nucleotide vs. Protein Dotplots

We observed low nucleotide identity through sparse dotplot relationships (Figure 2) and ANI values (Appendix A) for WhyPhy and Karezi, in contrast to evident conservation of genome architecture and synteny revealed through genome maps (Figure 4, Appendix A). We reexamined the relationships between these phages through a dotplot of concatenated protein sequences, which revealed considerable regions of amino acid identity across the width of these genomes (Figure 5), derived from various *Bacillus* species hosts. As these relationships are weakly observed at the genome level, incorporation of multiple protein sequence-based measures of similarity is essential to understanding phage genome evolution and taxonomy. Together, protein content analysis, examination of genome maps, and protein sequence dotplots revealed considerable conservation of genome structure, synteny, and protein function that are not evident at the genome level.

### 3.7. Endolysin and Holin

Putative endolysin and holin genes were identified and characterized using bioinformatics tools to examine the evolutionary relationships of these proteins between diverse *Bacillus* phages. The endolysin and holin genes in our podoviral genomes are found in tandem, as a canonical lysis “cassette”. Endolysin proteins are modular and contain two functionally-distinct domains: an N-terminal enzymatically active domain (EAD) and a C-terminal cell-wall binding domain (CBD) [54] connected by a flexible linker. The sequences of these domains were examined separately to compare domain architecture and phylogeny. This set of phages contains three distinct EADs and four distinct CBDs (Figure 6), with domains that functionally overlap endolysins that were characterized previously from a collection of *Bacillus*-infecting myoviruses [55]. This is the only protein with sequence homology that is conserved between podoviral and myoviral genomes, and these sequences contain functional domains that likely define specific interactions with the cell wall components of several *Bacillus* species.

Holin proteins were identified through identification of conserved domains, evaluation of sequences for transmembrane helix (TMH) domains, and colocalization in the genome with endolysin consistent with the gene arrangement in phi29, GA-1 [56,57]. Phylogenetic analysis (Figure 7) of holin protein sequences reveals that they are highly conserved within clades. The proteins are predicted to form 1 to 3 TMH domains, consistent with the wide diversity that is seen within the holin protein family [58]. We also examined holin sequences for the presence of features including a stem loop, two ribosome binding site/start codon positions, and two start codon methionines surrounding a positively charged amino acid, which might support a dual start translation mechanism identified in phage Lambda [59]. Similarly, the holin gene 14 of phage phi29 contains a unique mechanism that allows translation of two proteins from different start positions, and start codon mutational studies revealed lysis phenotypes supporting opposing roles in lysis timing for the two forms of the protein [56,57]. This dual start motif was first identified in phage Lambda, where the longer protein (S107) has an additional Met/Lys (MK) at the beginning of the protein and inhibits lysis while the shorter protein (S105) allows hole formation in the membrane. In phi29 the shorter holin (129 amino acids) causes rapid lysis and is more lethal to host cells than the longer form of holin (131 amino acids, [57]). The Whiting and Arbo1 genomes are identical to phi29 in ribosome binding site predictions and potential stem loop sequences. Notably, despite the high sequence conservation between phi29, Whiting18, and Arbo1 holin proteins, the phi29 MKM motif is instead MTM in Whiting18 and Arbo1, with a change of lysine to threonine in our sequences. To our knowledge, it has not been confirmed that a holin with a nonpolar residue between the two methionines functions as a dual start motif to control lysis timing. 

### 3.8. pRNA Prediction

A small RNA molecule that acts during genome packaging, the “pRNA”, has been experimentally studied in phi29 [60] and identified in a few other *Bacillus*-infecting podoviruses including GA-1 [61]. More recently, computational tools have been used to predict pRNA locations for some of the phages described in this article [62]. The pRNAs for Ademby, StevenHerd11, Thornton, Karezi, WhyPhy, Whiting18, and Arbo1 were predicted for this study based on homology, location, identification of nearby promoter/terminator, and maintenance of conserved residues (Appendix A, and [28]). The pRNA sequences for Ademby, StevenHerd11, and Thornton were predicted by homology to previously predicted pRNAs [28], and confirmation of the presence of transcription promoters as well as maintenance of some conserved residues despite the low sequence identity between these pRNAs and phi29 or GA-1. Transcription terminators were not identified for this set of pRNAs, presumably due to their location near the genome end. A pRNA sequence for WhyPhy was predicted through identification of a small 15 nucleotide (nt) region of identity between Stitch pRNA and WhyPhy genome, in a region with a nearby promoter proximal to the genome end. WhyPhy length (189 nt) was set according to alignment with phi29 and GA-1 and needs experimental verification. A pRNA sequence for Karezi was predicted through alignment of a core 49 nt pRNA region with 80% identity to GA-1, and confirmation of a nearby transcription promoter and terminator on the same strand as the pRNA. The pRNA for Karezi (and GA-1) is the only pRNA in our group that is not located at a genome terminus, and this intergenic region was confirmed to lack coding potential. Karezi pRNA ends were set through alignment with GA-1, for a length of 165 nt. Finally, the pRNA sequences for Whiting18 (179–352) and Arbo1 (317–174) were predicted directly through sequence alignment, as these pRNAs retained 99% identity to the phi29 pRNA sequence. Finally, RNAFold was used to examine potential conservation of pRNA tertiary structure of Domain 1, and to highlight the position of potential conserved residues identified experimentally [29,61] (Appendix A). Manual inspection of predicted tertiary structures confirmed the presence of potential conserved nucleotides in the CE loop and E helix [28] in Karezi and WhyPhy, lending support to these predictions. Harambe and SerPounce structures and sequences were more divergent, and identification of these residues was less clear. 

### 3.9. Sequence-Directed Recombination of Capsid and GNAT Family Acetyltransferase Genes

The genomes of SerPounce and Claudi, as well as the related genome Baseball_field, encode capsid proteins that are ~100 residues shorter than their orthologs. Alignment of the genome region corresponding to the capsid protein of KonjoTrouble and Claudi reveals a 6 bp repeat (5′-AAAAGG-3′) present on either side of an inserted 232 bp sequence (Figure 8a,b). The insertion appends a stop codon that allows translation to halt after completion of the essential, canonical HK97-like domain. The 232 bp insertion is nearly identical in SerPounce, Claudi, and Baseball_field, and reveals 78% query coverage and 78% identity to *B. cereus* group sequences in GenBank. In all three phage genomes, the sequence of the second domain of the capsid protein, a group 2 bacterial immunoglobulin-like domain (BIG2), continues after the 232 bp insertion, but without a start codon, and is presumed to not be translated. This domain is dispensable in phi29 and is not part of the canonical HK97 fold of bacteriophage capsid proteins [50]. The BIG2-like domain interacts with gp 8.5 head fiber protein [50]. Note, the gp 8.5 head fiber protein is also dispensable [63] and the *Claudivirus* genus phages appear to lack a homolog to phi29 gp 8.5 head fiber protein suggesting we are observing loss of these interacting yet dispensable components of the mature capsid.

The 3′-end of the *Claudivirus* genus phages appear to be a site where gene diversity is introduced. For example, the last gene in Thornton (gp 43) and SerPounce is functionally annotated as a GNAT family acetyltransferase. These proteins exhibit homology to *Escherichia* phage Mu methylcarbamoylase mom, which is predicted to modify ~15% of adenine residues in viral DNA to make it resistant to type 1/II restriction enzymes [64]. A region containing 21 bp of identity was identified on both sides of this gene in the Thornton genome (Figure 8c,d), compared to a single copy of this region in Juan. This exemplifies the use of sequence-directed recombination to expand the protein function capacity of these genomes and to provide phage with increased metabolic advantages. 

### 3.10. Non-Sequence-Directed Recombination Leads to Modifications to Essential DNA Polymerase

The DNA polymerase in phi29 is essential for phage replication and has been well- characterized as a B-family replicative polymerase [65]. This DNA polymerase is encoded as a single subunit of approximately 691 amino acids, and initiates replication using the terminal protein as a primer. The sequence encodes an N-terminal exonuclease domain and a C-terminal polymerase domain. Amino acid identity for this protein ranges from 39–100% identity within the genomes in our study. The DNA polymerase gene structure in Aurora represents a unique case. It is present in two separate predicted open reading frames arising by mutations within a 44 bp region between the two ORFs (gp 12 and gp 11, Figure 9A). The region begins with one “T” residue to create a stop codon that truncates translation after 480 residues. The mutation region ends with a second ORF, in the same translation frame as the first ORF, containing the final 183 residues of the protein. Overall genome sequence identity between the DNA polymerase region in Aurora and Juan is high quality without gaps, while identity in the 44 bp region is lower (73% with 18 gaps). The Aurora sequence contains no apparent signature indicating a sequence-dependent recombination event.

Interestingly, this break in Aurora DNA polymerase ORFs occurs in a region where the DNA polymerase sequences encode a 17 amino acid region that is present in about half of our sequences (Figure 9B). This region in phi29 contains the TPR2 domain [66], which interacts with the terminal protein to provide processivity and strand displacement function [65]. Mutation of this domain results in a DNA polymerase with lower processivity that is unable to perform strand displacement. Presumably, the variations of this domain of this group of DNA polymerases allow a reasonable level of DNA polymerization to occur. Given the essential characteristic of this protein, the Aurora DNA polymerase may be translated as one protein or functional as a two-subunit protein. Finally, the *Claudivirus* genus genomes contain a terminal protein family that is unrelated in sequence to the terminal protein family of phi29-like phages, perhaps to accommodate terminal protein interactions with the different TPR2 subdomain sequences in these genomes.

## 4. Discussion

The field of bacteriophage genomics has come a long way from classification of phages by morphology and establishment of the class *Caudoviricetes* for all tailed phages [67]. The “phi29 family” of *Bacillus*-infecting podoviruses were originally organized into three groups based on a variety of phenotypic and genome data of phages using *B. subtilis* as a host: group I containing phi29/PZA/phi15/BS32; group II including B103/Nf/M2Y; and group III having GA-1 as the sole member [68]. The phage phi29 was identified as a type species in International Committee on Taxonomy of Viruses (ICTV) morphology-based taxonomy. Sequencing of phage genomes has revealed remarkable genomic diversity that was not accounted for by phenotypic taxonomy. Recently, the ICTV released an updated taxonomic framework that emphasizes comparative sequence analysis and includes up to 15 taxonomic ranks [12]. The number of phages with sequenced genomes affiliated with the original phi29 family group is now much expanded. Together the genomes described here fall into five different genera within the *Salasmaviridae* family in the updated ICTV virus taxonomy.

This diverse group of phages was derived from multiple *Bacillus* hosts. The phage genomes range from 18,548–27,206 bp in length, 30.1–39.6% GC content, and contain 25–46 predicted open reading frames. The smallest genomes are constructed with a higher genetic efficiency, with ~93% of the genome corresponding to open reading frames for Arbo1 compared to ~83% of the genome for our largest phage SerPounce. Grouping of genomes corresponds to prior organizational structures [42], and shows that while genome sequences between groups can be quite diverse, remarkable levels of genome architecture and protein content are conserved across these small viruses. A conserved “Core Genome” of 31–33 genes for T4-related bacteriophages has been described [69], and presumed to be the most ancient portion of these genomes. The conserved set of at least fifteen protein functions across these small *Bacillus* phages from different hosts represents ~78% of the genome for our smallest genomes and ~56% of the genome for the largest genome. When compared to phi29, GA-1, and MG-B1, the presence of these protein functions within a diverse set of genomes discovered over decades, as well as geography, implies this set of functions represents a set of core requirements for productive infection of *Bacillus*, with a sequence diversity that also suggests ancient origins.

Sequence-directed recombination events within these conserved protein functions were identified in the DNA polymerase and capsid genes. We observed evidence of deletion of a small domain extraneous to the core function of each of these essential proteins. These recombination events are highly precise in nature, leaving most of these essential proteins intact. It is presumed these deletions have minimal negative impact on protein function, highlighting the malleability of individual essential genes in this system that support productive phage infection.

While the order of these core conserved genes is maintained, genome mosaicism is observed consistent with previous findings with *Bacillus* and other phages [6,7,10,42,70]. Phages from *B. thuringiensis kurstaki* hosts have the largest genomes in this study and highlight the ability of phages to continuously acquire new genes [7]. The center and terminal areas of these genomes appear to be more frequent sites for insertion of additional, novel protein-coding genes, in contrast to the structural gene region where mosaicism is rarely observed, consistent with observations from mycobacteriophages [6]. Regions of genome mosaicism within these genomes expand significantly on *Bacillus* podoviral protein diversity, with the addition of a combined 19 novel genes that lack homologs in GenBank and therefore have an undefined evolutionary history. Our 16 phage dataset, as part of a custom Phamerator database with 265 *Bacillus* phage genomes, has 71 orphams, representing 12.5% of the total number of proteins in this study. Hatfull et al. suggest protein phams that are present in a single genome are likely candidates for recent acquisition as opposed to loss, and it is thought these insertions may occur through an illegitimate recombination mechanism [6]. A larger study of 627 mycobacteriophage genomes revealed phage genomes experience a constant influx of genes from other organisms [7]. This combined set of uncharacterized proteins may represent specific adaptations of these phages in nature that may be resolved as the collection of sequenced *Bacillus* phages is expanded.

Finally, while this gene population appears to be relatively genetically isolated, one protein family extended across a broader range of *Bacillus* phages. Bacteriophage endolysin proteins are known for their modular architecture, including an N-terminal enzymatically active domain (EAD) and C-terminal cell-wall binding domain (CBD) connected by a short linker [71,72]. Phylogenetic analysis of the separate EAD and CBD compared to functionally characterized domains indicates the presence of EAD and CBD in genomes of otherwise unrelated phages from podoviral and myoviral morphology groupings. For example, PlyB endolysin is from phage Bcp1, a myoviruse isolated with *B. anthracis* as a host bacteria, and the PlyB EAD exhibits broad activity against *B. cereus* group bacteria [73]. This EAD sequence maintains ~72% identity to the PlyB-like EADs present in our podoviral phages isolated with *B. thuringiensis kurstaki* as a host bacteria. The endolysin EADs of *B. pumilus* phages Taylor, WhyPhy, and Karezi maintain ~40–48% identity for the domain, predicted to be an N-acetylmuramoyl-L-alanine amidase. While these phages were isolated using the same host bacteria system, they exhibit different phage particle morphology, genome size, and content. Because the CBD is thought to confer host specificity, it is perhaps most surprising that the SH3 CBD from *B. cereus* phage B4 maintains 62% identity with the CBD from *B. thuringiensis kurstaki* phages BeachBum and Harambe, again phages of different morphology, genome size, gene content, and host systems. Together, these comparisons indicate the modular composition and functional variability of phage endolysins extends broadly across typical host and morphology boundaries.

## Figures and Tables

**Figure 1 viruses-15-00276-f001:**
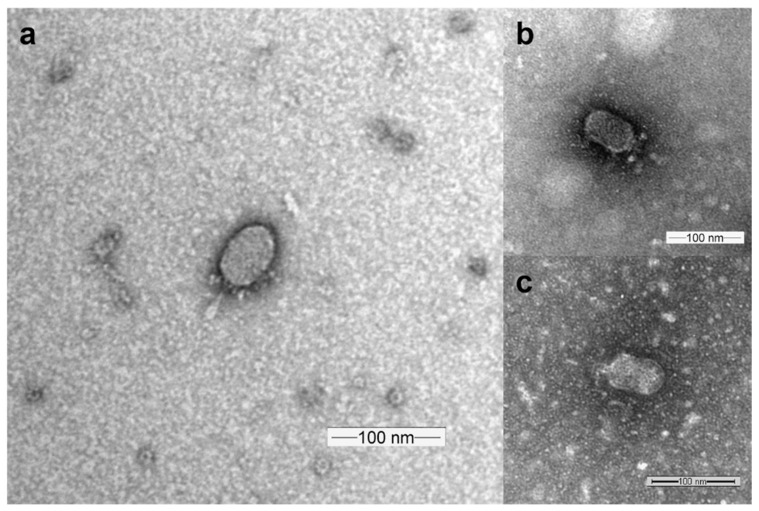
Representative transmission electron microscopy images for SerPounce (**a**), Beachbum (**b**) and Harambe (**c**).

**Figure 2 viruses-15-00276-f002:**
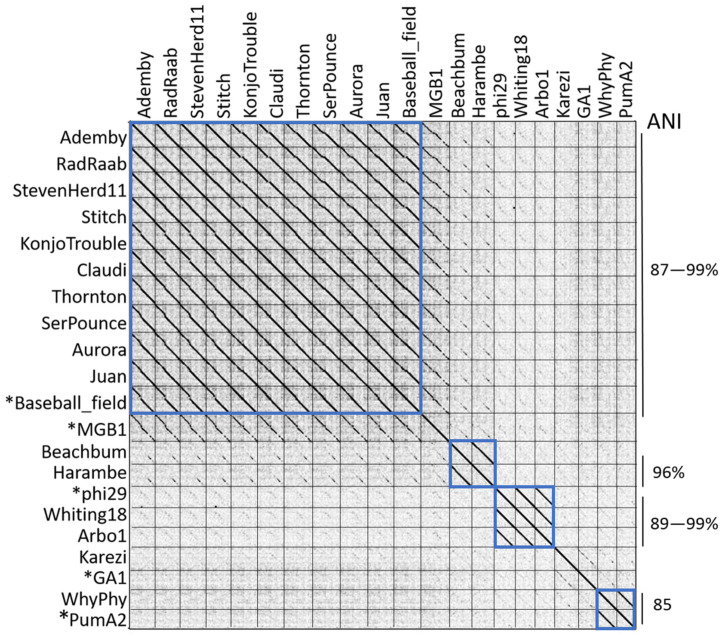
Whole genome sequence comparison by dotplot. Concatenated whole genome sequences were compared to visualize genome identity by dotplot. Blue boxes represent genome groupings. Lines on the right indicate groupings of similar genomes labeled with average nucleotide identity values. An asterisk indicates genomes obtained from GenBank.

**Figure 3 viruses-15-00276-f003:**
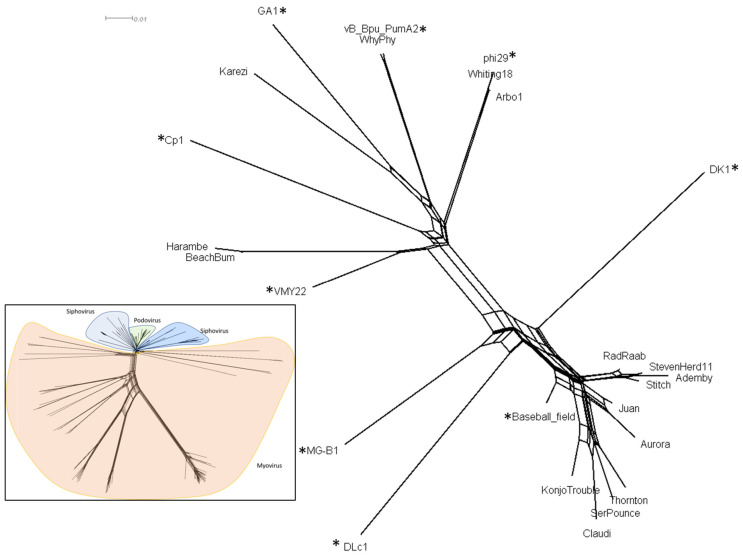
Protein content comparison of representative *Salasmaviridae* bacteriophage genomes. A profile of protein content extracted from a custom Phamerator database was visualized using Splitstree v4.0. An asterisk was used to designate phages that were added to our dataset for comparative purposes. Inset shows podovirus are separate from myovirus and siphovirus, from a database of 265 *Bacillus* phage genomes.

**Figure 4 viruses-15-00276-f004:**
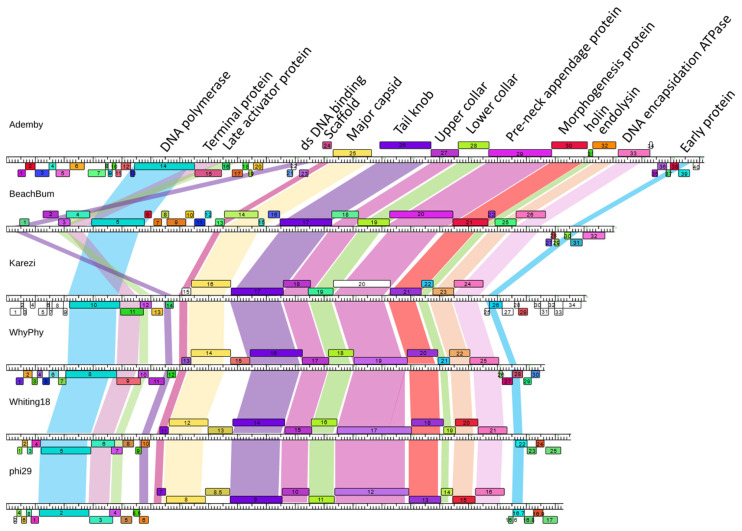
Maintenance of synteny and function in representative genomes. The regions between fifteen protein families were shaded according to the color of the protein pham in Ademby to highlight conservation of protein function throughout these diverse phage genomes. A full collection of Phamerator maps is available in Appendix A.

**Figure 5 viruses-15-00276-f005:**
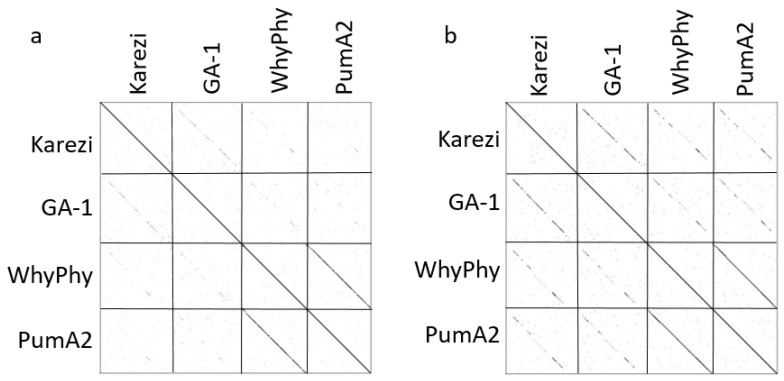
Whole genome and proteome comparison by dotplot. Whole genome (panel **a**, word size = 12) and whole proteome (panel **b**, word size = 4) sequences are compared against themselves.

**Figure 6 viruses-15-00276-f006:**
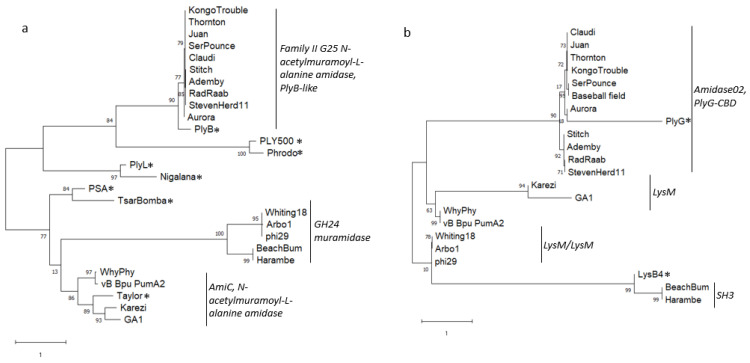
Phylogenetic analysis of endolysin protein sequences. Protein sequences were analyzed separately by (**a**) enzymatically active domain (EAD) and (**b**) cell-wall binding domain (CWB)The trees with the highest log likelihood are shown. The tree is drawn to scale, with branch lengths measured in the number of substitutions per site. Bootstrap values indicate the percent of iterations producing replicate clade arrangements after 500 iterations. This analysis involved 26 (EAD) and 22 (CBD) amino acid sequences. There were a total of 208 (EAD) and 123 (CBD) positions in the final dataset. Asterisk (*) symbols are used to highlight endolysin sequences from myoviral genomes.

**Figure 7 viruses-15-00276-f007:**
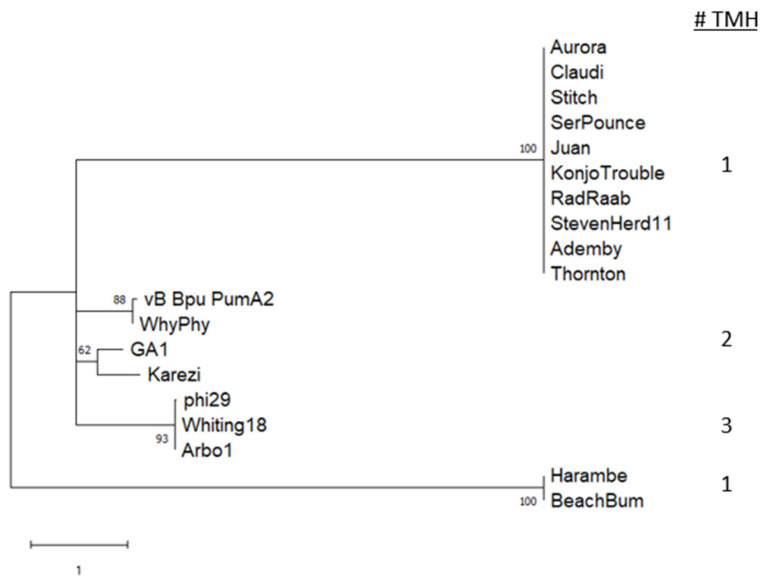
Holin phylogeny. The tree with the highest log likelihood (−1616.44) is shown. The tree is drawn to scale, with branch lengths measured in the number of substitutions per site. Bootstrap values indicate the percent of iterations producing replicate clade arrangements after 500 iterations. This analysis involved 19 amino acid sequences. There were a total of 139 positions in the final dataset. The number of predicted transmembrane helices (TMH) are listed in the right column.

**Figure 8 viruses-15-00276-f008:**
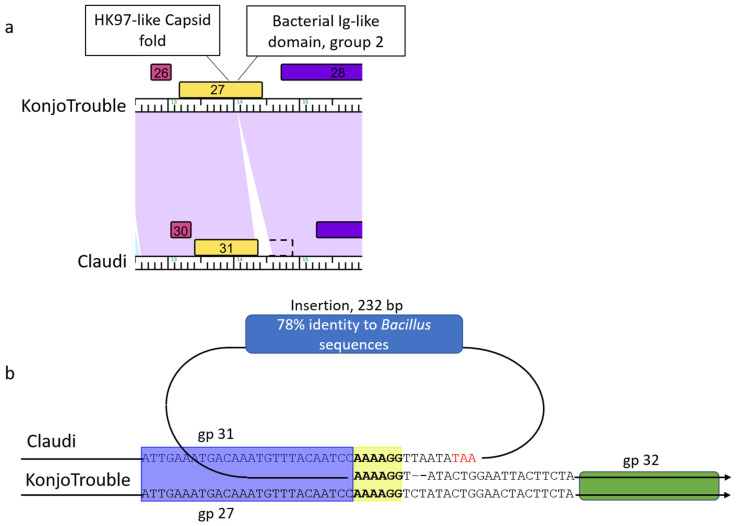
Sequence-directed recombination leads to protein variations and new protein functions. (**a**) Phamerator map of the capsid protein region of KonjoTrouble compared to Claudi. Colored rectangles represent protein families and purple shading between the genomes represents Blastn homology. The dotted black outline represents where the BIG2 domain is present in Claudi sequence. (**b**) Sequence alignment of the region surrounding the 232 bp insertion in Claudi compared to KonjoTrouble, which lacks this insertion. Blue and green rectangles are colored corresponding to protein family coloring in panel (**a**). The yellow shaded region shows two copies of a 6 nucleotide repeat in Claudi, and one copy in KonjoTrouble. The red TAA designates a stop codon in the Claudi sequence. (**c**) Phamerator map of the 3′-end region of Juan compared to Thornton. (**d**) Sequence alignment of the region surrounding Thornton gp 43.

**Figure 9 viruses-15-00276-f009:**
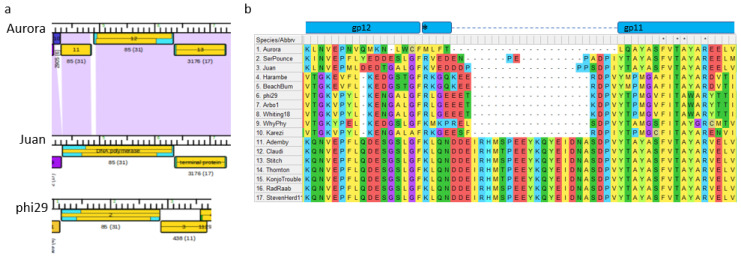
Mutation in DNA polymerase sequence reveals variable region in essential enzyme. (**a**) Phamerator map of the DNA polymerase protein region of Aurora compared to Juan and phi29. Colored rectangles represent protein families and purple shading between the genomes represents Blastn homology. (**b**) Multiple sequence alignment of the deletion region in Aurora reveals this sequence is also missing in other DNA polymerase proteins. The asterisk indicates the start methionine for the second region encoding Aurora DNA polymerase (gp 11 in panel **a**).

**Table 1 viruses-15-00276-t001:** Genome features of 16 *Bacillus* phages.

Phage Name	Genome Length (bp)	GC%	Number of Genes	Host	GenBank ID	Inverted Repeat Ends
Genus: *Claudivirus*						
Ademby	24,162	30.7	40	Btk *	OL744112.1	ND
RadRaab	23,946	30.6	34	Btk	MF156580.1	8 bp ITR, 5′-AAATGTAA
StevenHerd11	23,953	30.6	37	Btk	MK084630.1	11 bp ITR, 5′-AAATGTAAGGG
Stitch	24,320	30.4	36	Btk	KX349901.1	7 bp ITR, 5′-AAATGTA
KonjoTrouble	26,061	30.1	40	Btk	MF156578.1	9 bp ITR, 5′-AAATGTAAA
Claudi	26,502	30.3	46	Btk	KX349900.2	8 bp ITR, 5′-AAATGTAA
Thornton	26,319	30.5	43	Btk	MW348917	8 bp ITR, 5′-AAATGTAA
SerPounce	27,206	30.4	44	Btk	KY947509	8 bp ITR, 5′-AAATGTAA
Aurora	25,905	30.6	40	Btk	KX349899.2	5′ end is 5′-AAATGTAA, 3′ end ND
Juan	25,032	30.6	34	Btk	MF156577.1	9 bp ITR, 5′-AAATGTAAA
Genus: *Harambevirus*						
BeachBum	21,054	35.4	30	Btk	KY921761.1	16 bp ITR, 5′-AAGATAGCCCCCCACC
Harambe	21,684	35.3	33	Btk	KY821088	16 bp ITR, 5′-AAGATAGCCCCCCACC
Genus: *Karezivirus*						
Karezi	20,083	37.3	34	*B. pum.** SAFR 32	MN082625	7 bp ITR, 5′- AAATTAG
Genus: *Bundooravirus*						
WhyPhy	18,642	34.9	28	*B. pum* SAFR 32	MW419775	12 bp ITR, 5′-AATGTAAAGGTA
Genus: *Salasvirus*						
Whiting18	19,548	39.6	25	*Bacillus* sp. 203	MW477480	ND
Arbo1	19,362	39.6	25	*Bacillus* sp. 203	OL744111.1	ND

* Btk = *Bacillus thuringiensis kurstaki*; B. pum. = *Bacillus pumilus;* ND = not determined; ITR = inverted terminal repeat.

## Data Availability

Genome reads have been archived through SRA submissions PRJNA748677 and PRJNA732421.

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
