# Peer review of "Evidence of a Set of Core-Function Genes in 16 Bacillus Podoviral Genomes with Considerable Genomic Diversity"

_viruses, 2023, doi:10.3390/v15020276_

Round 1

Reviewer 1 Report

Comments

This work is a comparative genomic study of Bacillus bacteriophages. I think the manuscript could benefit if the authors present their results with a broader approach. The comparative genomic analysis should be put in the context of the Salasmaviridae family of bacteriophages. Also if available more sequences should be included in datasets used for phylogenetic and splitstree analyses. 

Abstract: 

- Line 13. Which is the taxonomic classification of this podoviruses? Podoviridae should be updated to the new taxonomic families.

Keywords:

- Podoviridae should be replaced by podoviruses

Introduction:

-In lines 39-42 and lines 55-57 the authors briefly comment aspects of the taxonomy of Bacillus phages. It will be helpful if they include a brief paragraph of the actual taxonomic classification, because it is not clear through the manuscript. Which families have been described for Bacillus phages and to which family or genus belong the bacteriophages described in this work? 

Materials and methods:

- In the section 2.1 Host bacteria and after. Instead of "discovered", "isolated" could be more suitable to describe bacteriophage isolation. 

- Parameters and criteria used in the comparative genome analyses in Section 2.5 should be further described. Which and how many sequences were included in datasets for each analysis? Could more sequences from databases be included? A brief description of the phylogenetic analyses is needed. The recombination analyses are not mentioned in this section, were there any formal analyses performed in order to find location of the recombination events? Which bioinformatic tools were used for recombinant analyses? Which was the criteria used to assure that a recombination event took place? 

Results:  

- In line 141 it is mentioned that "This group of phages are phi29-like Caudoviricetes ...". Do the bacteriophages isolated in this work belong to Salasvirus phi29 species, or are a different species within the genus Salasvirus, or even a different genus of the Salasmaviridae family? The belonging of the isolated phages to Family Salasmaviridae is first mentioned in line 220,section 3.5.

By searching to which genus Baseball-field, and PumA2 belong I could interpretate that Ademby group belong to Northropvirinae subfamily and Claudivirus genus, WhyPhy is a Bundooravirus, and Whiling18 and Arbo1 are Salasvirus. Is this correct? Why is not mentioned through the manuscript? Is there any uncertainty regarding their taxonomic classification?

- In section 3.2. Genome relationships to sequenced bacteriophages: The word homolog is repeatedly used to refer to related genomes available in GenBank database. The use of homolog is not correct since similarity and homology are not equivalent. Homolog could be replaced with another more suitable term, perhaps "related genomes"? 

- In section 3.2. I found this section not very informative, the relationships described there are also discussed in the following sections with better support. 

- Lines 190-192: "A small shaded area observed on the diagonal for Harambe and BeachBum contains a set of conserved proteins found in almost all of our genomes" Which functions are present in this set of conserved proteins? Are the 15 same genes mentioned before for all bacteriophages in analyses? Are they orthologues? could them be used to perform aditional phylogenetic analyses? 

- Which is the %ANI between BeachBum-Harambee and Karezi and their most closely related bacteriophage within the Salasmaviridae family? is 59% with MGB1 and 64 with GA-1? Could they be a new genus of the Salasmaviridae family? I think the authors should include at least one genome from every genus of the Salasmaviridae family to the %ANI evaluation. 

- Lines 219-221, "Three genomes were added to our analysis to represent one member of each genus within the family Salasmaviridae: Cp1 (Z47794.1, [43]) , DLc1 (MW012634.1, [44]) and VMY22 (KT780304.1, [45])". Looking at the ICTV it shows that Salasmaviridae, has 3 subfamilies and 12 genus. Why only these 3 genomes selected? Why not include at least one genome from every genus of the Salasmaviridae? If I am not mistaken 8 of the 12 genus are included in the analyses. 

- In figure 3, it would help if the authors point out groups of phages with their classification and highlight the sequences from this study.

- Lines 247-250: “The Ademby group shares a high level of genome sequence similarity as well as protein content, and a region where the direction of transcription switches in the middle of the

genomes that shows expansion of diversity of proteins by the presence of novel genes.” 

In Figure 4, it is shown that Karezi, WhyPhi, Whiting18 and phi29 also have this transcriptional organization so how the authors explain the statement in lines 247-250?

- Lines 272-274: "manual inspection of this set of 16 genomes reveals each contains a core set of 15 protein functions (Figure 4) and maintains a remarkable conservation of synteny and function despite low nucleotide identity relationships" Are the protein sequences of these genes conserved? May be the authors can perform phylogenetic analyses of proteins like DNA polymerase, major head protein, tail knob protein, upper collar protein, and DNA encapsidation ATPase to analyze the phylogenetic relationships of the Salasmaviridae family.

- Figures 3 and 4, the  description of the phylogenetic analyses shown with the figures should be placed in Materials and Methods. In line 336- 337: “The percentage of trees in which the associated taxa clustered together is shown next to the branches.” I think it is better to indicate the method used for support estimation, was it bootstrap?

- Lines 348-350: "Phylogenetic analysis (Figure 7) of holin protein sequences reveals that they are highly conserved within branches and are predicted to form a variety of TMH domains" What does "conserved within branches" mean in this phrase? Do you mean conserved within a clade? Also the prediction os THM domains is not a result of a phylogenetic analysis. Please rephrase this statement. 

Section 3.10 and 3.11, recombination events described in these sections are just visual observations? did the authors tried to detect recombination evidence with any bioinformatic software?

Discussion:

- Line 493, Caudovirales order is now a class Caudoviricetes. 

- Lines 492-505. This paragraph should be updated in order to briefly described the current classification of these bacillus phages. 

- Lines 504-505: The authors say: "Together the genomes described here fall into 5 different genera within the Salasmaviridae family in the updated ICTV virus taxonomy". As mentioned before, this is not described in the results section and it will be really helpful to do so.

- Lines 531-533: "Certain areas of these genomes appear to be more frequent sites for insertion of additional protein coding genes, perhaps through an illegitimate recombination mechanism[6] and on a relatively recent timeline". Which were these certain areas? How do the authors established the timeline?

Author Response

This work is a comparative genomic study of Bacillus bacteriophages. I think the manuscript could benefit if the authors present their results with a broader approach. The comparative genomic analysis should be put in the context of the Salasmaviridae family of bacteriophages. Also if available more sequences should be included in datasets used for phylogenetic and splitstree analyses. 

We thank the reviewer for encouraging a broader approach connected to the current ICTV taxonomy. We have incorporated improvements throughout the manuscript to make these connections. Importantly, we believe Streptococcus phage Cp1 and Cp7 should be removed from the Salasmaviridae family based on some updated analysis suggested by this reviewer, and will refer that suggestion to the appropriate study committee. We appreciate your careful reading and suggestions. Responses to specific comments are detailed below.

Abstract: 

- Line 13. Which is the taxonomic classification of this podoviruses? Podoviridae should be updated to the new taxonomic families.

 “Podoviridae” was updated to “in the Salasmaviridae family”.

Keywords:

- Podoviridae should be replaced by podoviruses

Done

 Introduction:

 -In lines 39-42 and lines 55-57 the authors briefly comment aspects of the taxonomy of Bacillus phages. It will be helpful if they include a brief paragraph of the actual taxonomic classification, because it is not clear through the manuscript. Which families have been described for Bacillus phages and to which family or genus belong the bacteriophages described in this work? 

As far as we are aware, Grose et al, published in 2014 , included all Bacillus phages downloadable from Genbank at the time. The phages in Grose et al., included a wide variety of genomes including many families within the ICTV taxonomy classes Tectiliviricetes and Caudoviricetes, with most of the phages being organized under the familes Herelleviridae and Spounaviridae. Several phages described in the paper are not included in ICTV taxonomy. The text was modified as follows: ‘including families within the classes Tectiliviricetes and Caudoviricetes, and eight Salasmaviridae family phages’ that were in Genbank at the time of publication.

Materials and methods:

 - In the section 2.1 Host bacteria and after. Instead of "discovered", "isolated" could be more suitable to describe bacteriophage isolation. 

 Done

- Parameters and criteria used in the comparative genome analyses in Section 2.5 should be further described. Which and how many sequences were included in datasets for each analysis?

This text was added to the comparative genomics methods section to summarize genomes that were analyzed: Sixteen genome sequences from phages isolated by us are listed in Table 1 with Genbank identifiers. These sequences were compared to five best BLASTN matches [Baseball_field (MT777452.1), MG-B1 (KC685370.1), GA-1 (X96987.2), PumA2 (MN524845.1) and phi29 (EU771092.1)] in dotplot analysis, as well as 5 additional select species to represent each genus in the family Salasmaviridae [DK1 (MK284526.1),  DLc1 (MW012634.1), Cp1 (AH001309.2), VMY22 (KT780304.1) and B103 (X99260.1)] in ANI and Splitstree analysis.

Could more sequences from databases be included?

Additional sequences to represent each genus within Salasmaviridae were added to ANI and splitstree analysis to address a results section comment below.

 A brief description of the phylogenetic analyses is needed.

Methods were moved from the figure 6 and 7 legend to the methods. The new text for each section is below:

Methods: Evolutionary analyses were conducted in MEGA X [37], and inferred by using the Maximum Likelihood method and JTT matrix-based model [53].  Initial tree(s) for the heuristic search were obtained automatically by applying Neighbor-Join and BioNJ algorithms to a matrix of pairwise distances estimated using the JTT model, and then selecting the topology with superior log likelihood value. Bootstrap iterations were set to 500.

Figure 6 legend: Phylogenetic analysis of endolysin protein sequences. The trees with the highest log likelihood are shown. The percentage of trees in which the associated taxa clustered together is shown next to the branches. The tree is drawn to scale, with branch lengths measured in the number of substitutions per site. Bootstrap values indicate the percent of iterations producing replicate clade arrangements after 500 iterations. This analysis involved 26 (EAD) and 22 (CBD) amino acid sequences. There were a total of 208 (EAD) and 123 (CBD) positions in the final dataset. Asterisk (*) symbols are used to highlight endolysin sequences from myoviral genomes.

Figure 7 legend: Holin phylogeny. The tree with the highest log likelihood (-1616.44) is shown. The tree is drawn to scale, with branch lengths measured in the number of substitutions per site. Bootstrap values indicate the percent of iterations producing replicate clade arrangements after 500 iterations. This analysis involved 19 amino acid sequences. There were a total of 139 positions in the final datasetThe number of predicted transmembrane helices (TMH) are listed in the right column

 The recombination analyses are not mentioned in this section, were there any formal analyses performed in order to find location of the recombination events? Which bioinformatic tools were used for recombinant analyses? Which was the criteria used to assure that a recombination event took place? 

This text was added to the comparative genomics methods: Recombination events were visually identified using pairwise comparisons in Phamerator to examine genomes for changes (protein length, genome alignment). Regions showing interesting differences were further analyzed using sequence alignment through BLASTN and ClustalOmega.

Results:  

 - In line 141 it is mentioned that "This group of phages are phi29-like Caudoviricetes ...". Do the bacteriophages isolated in this work belong to Salasvirus phi29 species, or are a different species within the genus Salasvirus, or even a different genus of the Salasmaviridae family?

These phages belong to a variety of genera within the Salasmaviridae family. I believe that is now clear due to updates throughout the manuscript. 

 The belonging of the isolated phages to Family Salasmaviridae is first mentioned in line 220,section 3.5.

The family Salasmaviridae is now mentioned in the abstract, three times in the introduction, etc. as a result of your suggestions. This first result paragraph is about particle morphology and genome sequence characteristics. Taxonomy relationships were added to the revised sections 3.2 and 3.3 below, where genome comparisons are introduced.

By searching to which genus Baseball-field, and PumA2 belong I could interpretate that Ademby group belong to Northropvirinae subfamily and Claudivirus genus, WhyPhy is a Bundooravirus, and Whiling18 and Arbo1 are Salasvirus. Is this correct? Why is not mentioned through the manuscript? Is there any uncertainty regarding their taxonomic classification?

Genus groups were not previously mentioned in our manuscript and have now been added to Table 1 as well as some results. The text “These genomes belong to the Salasmaviridae family, and blue boxes were used to highlight likely genus groups within the sequences we compared (Figure 2)” has been added to section 3.2, line 205, since we don’t determine those groupings. We do suggest Streptococcus phages Cp1 and Cp7 be removed from the Salasmaviridae family, and will refer this to the appropriate study group.

- In section 3.2. Genome relationships to sequenced bacteriophages: The word homolog is repeatedly used to refer to related genomes available in GenBank database. The use of homolog is not correct since similarity and homology are not equivalent. Homolog could be replaced with another more suitable term, perhaps "related genomes"? 

 The term homolog was replaced with related genomes as suggested.

- In section 3.2. I found this section not very informative, the relationships described there are also discussed in the following sections with better support. 

Sections 3.2 and 3.3 were combined into one section describing query coverage and dot plot visualization to condense this portion of the text, and to include a taxonomic grouping perspective.

- Lines 190-192: "A small shaded area observed on the diagonal for Harambe and BeachBum contains a set of conserved proteins found in almost all of our genomes" Which functions are present in this set of conserved proteins? Are the 15 same genes mentioned before for all bacteriophages in analyses? Are they orthologues? could them be used to perform aditional phylogenetic analyses? 

 That sentence referenced above ended up getting deleted from that section when we combined sections 3.2 and 3.3. The questions you are asking are discussed later, in section 3.4 Protein Content and 3.5 Phamerator Maps (new section numbers), including the set of functions that are conserved in all of the phages. Yes these genes are considered orthologues, and functionally similar. We didn’t incorporate additional phylogenetic analysis into this manuscript, as two of those genes are used in routine ICTV classification and full phylogenetic analysis of those families may be beyond the scope of this work.

- Which is the %ANI between BeachBum-Harambee and Karezi and their most closely related bacteriophage within the Salasmaviridae family? is 59% with MGB1 and 64 with GA-1? 

Yes

Could they be a new genus of the Salasmaviridae family?

Currently, MG-B1 and BeachBum/Harambe are two separate genera, and GA-1 and Karezi are separate genera, all within the Salasmaviridae family. GA-1 and Karezi are grouped within the Subfamily Tatarstanvirinae.

 I think the authors should include at least one genome from every genus of the Salasmaviridae family to the %ANI evaluation. 

Done, see Figure 3 (splitstree) and the updated Supplemental Table 1.

 - Lines 219-221, "Three genomes were added to our analysis to represent one member of each genus within the family Salasmaviridae: Cp1 (Z47794.1, [43]) , DLc1 (MW012634.1, [44]) and VMY22 (KT780304.1, [45])". Looking at the ICTV it shows that Salasmaviridae, has 3 subfamilies and 12 genus. Why only these 3 genomes selected? Why not include at least one genome from every genus of the Salasmaviridae? If I am not mistaken 8 of the 12 genus are included in the analyses. 

The ANI and Splitstree were updated with DK1 and B103 to cover all 12 genera within Salasmaviridae.

- In figure 3, it would help if the authors point out groups of phages with their classification and highlight the sequences from this study.

An asterisk was added to the image to note phages that were added to our dataset for comparative purposes.

- Lines 247-250: “The Ademby group shares a high level of genome sequence similarity as well as protein content, and a region where the direction of transcription switches in the middle of the

genomes that shows expansion of diversity of proteins by the presence of novel genes.” 

In Figure 4, it is shown that Karezi, WhyPhi, Whiting18 and phi29 also have this transcriptional organization so how the authors explain the statement in lines 247-250?

The statement was meant to describe the Ademby group, but doesn’t mean those 3 characteristics don’t apply to other groups. This sentence was added at line 305 to clarify “WhyPhy, Whiting18, Arbo1 and Karezi exhibit transcriptional organization similar to the Ademby group, in contrast to BeachBum and Harambe.”

- Lines 272-274: "manual inspection of this set of 16 genomes reveals each contains a core set of 15 protein functions (Figure 4) and maintains a remarkable conservation of synteny and function despite low nucleotide identity relationships" Are the protein sequences of these genes conserved? May be the authors can perform phylogenetic analyses of proteins like DNA polymerase, major head protein, tail knob protein, upper collar protein, and DNA encapsidation ATPase to analyze the phylogenetic relationships of the Salasmaviridae family.

We have explored these protein families to show that five protein families are highly conserved, and two of these families are already used routinely for phylogeny as part of ICTV taxonomy analysis. Further phylogenetic analysis of these families would likely be beyond the scope of this paper. 

- Figures 3 and 4, the  description of the phylogenetic analyses shown with the figures should be placed in Materials and Methods. In line 336- 337: “The percentage of trees in which the associated taxa clustered together is shown next to the branches.” I think it is better to indicate the method used for support estimation, was it bootstrap?

 Done, and bootstrap detail was added.

 - Lines 348-350: "Phylogenetic analysis (Figure 7) of holin protein sequences reveals that they are highly conserved within branches and are predicted to form a variety of TMH domains" What does "conserved within branches" mean in this phrase? Do you mean conserved within a clade?

“Branches” has been replaced by “clade”.

Also the prediction os THM domains is not a result of a phylogenetic analysis. Please rephrase this statement. 

Done, the TMH description has been separated from the phylogeny results.

Section 3.10 and 3.11, recombination events described in these sections are just visual observations? did the authors tried to detect recombination evidence with any bioinformatic software?

Phamerator software was used to identify the described recombination events, as blastn shading between the genomes and differences in protein lengths are readily evident in a pairwise comparison. This has been added to the methods.

Discussion:

- Line 493, Caudovirales order is now a class Caudoviricetes. 

Fixed

- Lines 492-505. This paragraph should be updated in order to briefly described the current classification of these bacillus phages. 

The purpose of the paragraph was to describe the historical shift from using morphology to genome sequences in the ICTV taxonomy. Genus names have been added to Table 1, the second results section (3.2) and language has been shifted throughout the manuscript to more strongly reflect this taxonomy. This particular paragraph hasn’t been updated as I am not sure what to add. I believe the reviewer was probably waiting to finally get to “5 different genera’ language that is in the paragraph, and that is now also reflected and expanded in multiple places in the manuscript prior to the discussion. We believe there is a stronger taxonomic representation throughout the manuscript, and thank the reviewer for this direction.

- Lines 504-505: The authors say: "Together the genomes described here fall into 5 different genera within the Salasmaviridae family in the updated ICTV virus taxonomy". As mentioned before, this is not described in the results section and it will be really helpful to do so.

We believe this has now been addressed through prior suggestions in the results section.

- Lines 531-533: "Certain areas of these genomes appear to be more frequent sites for insertion of additional protein coding genes, perhaps through an illegitimate recombination mechanism[6] and on a relatively recent timeline". Which were these certain areas? How do the authors established the timeline?

This paragraph has been modified and rearranged for clarity, including adding two references to support the concepts of mosiacism and acquisition of novel genes. While these events are presumed to occur on a more recent timeline compared to the older origins of shared conserved regions, we didn’t do specific analysis to confirm this so have removed the timeline phrase from the paragraph.

Thank you again for your excellent suggestions and attention to improve this manuscript.

Reviewer 2 Report

The work presented by Ahmed Ismail and co-workers describes the characteristic of newly isolated phages against the genus Bacillus including in-depth genomic characteristics. Results provide much valuable information for the phage community.

The introduction properly situates the subject of study, and the material and methods are adequately documented. Results provide many details and are correctly exposed.

Minor comments:

Line 13, 42: podoviridae – it’s the name of the virus family. 1) it should be written in italics and started with a capital letter; 2) change podoviridae to Podoviridae phages or phages from the Podoviridae family

Line 40: proper citation

Line 44: Enterobacteriophages change to Enterobacteriaceae phages

Line 50: the full name of Bacillus subtilis should be given

Line 58: remove the sentence „We’ve engaged students in phage hunting using Bacillus species as host bacteria for several years, due to the ubiquitous nature of Bacillus in the environment as well as the ease of propagation on simple media.”

Line 83: Change Discovery to Isolation

Line 84: there is not matter if students or researchers took samples, especially if they are co-authors or contributors to this paper.

Line 86: describe the phage isolation method with details

Line 112: remove students

Line 134: Change Discovery to Isolation

Line 135 – 138: Rewrite the sentence with no information about students or Undergraduate researchers

Line 141: If all characterized phages are the same morphology please add a table with capsids and tails size to supplementary materials and additional information within the text that in Figure 1 are representative morphologies.

Line 493: Caudovirales should be italicized

Line 516: Bacillus should be italicized

Author Response

The work presented by Ahmed Ismail and co-workers describes the characteristic of newly isolated phages against the genus Bacillus including in-depth genomic characteristics. Results provide much valuable information for the phage community.

The introduction properly situates the subject of study, and the material and methods are adequately documented. Results provide many details and are correctly exposed.

We thank the reviewer for their comments and appreciate their careful reading and suggestions. We want to note the four suggestions to remove mention of student involvement were not incorporated into the revisited manuscript. While the language may seem unusual to this reviewer, we do this work within a nationwide undergraduate research course (SEA PHAGES) and would like to highlight the direct involvement of a large number of students in this work. It is important for the context of SEA PHAGES that we are able to talk and write about the important contributions of undergraduate researchers in our publications, and to share papers containing language like this with our students. The 3 student coauthors listed on this manuscript completed analysis that was beyond the scope of our research based courses, but this analysis could not have been done without that prior work by 213 students. We are proud of the contributions they have made, would like to communicate their involvement, and hope this mindset is respected. 

Further responses to specific comments are detailed below. We again thank the reviewer for their suggestions to improve the manuscript.

Minor comments:

Line 13, 42: podoviridae – it’s the name of the virus family. 1) it should be written in italics and started with a capital letter; 2) change podoviridae to Podoviridae phages or phages from the Podoviridae family

As these family names have been removed from ICTV taxonomy, we were using them as a morphology designation. Line 13 was edited to read podoviruses (morphology) for a keyword, and line 42 was changed to Salasmaviridae family for the taxonomic designation. There are now many updates to include the proper ICTV taxonomy designations throughout the manuscript.

Line 40: proper citation

I believe the issue was with location, and moved the citation placement from the end of the sentence to after the Author name, I hope this is sufficient?

Line 44: Enterobacteriophages change to Enterobacteriaceae phages

Done

Line 50: the full name of Bacillus subtilis should be given

The name is fully written out at first usage.

Line 83: Change Discovery to Isolation

Done

Line 86: describe the phage isolation method with details

The phage isolation methods were expanded to include some more detail: “Phage populations were purified by standard SEA-PHAGES methods[14] through at least three rounds of plaque purification by picking a well-isolated plaque into phage buffer, serially diluting, infecting into culture aliquots and plating with top agar. After purification, a plaque was picked into phage buffer and the sample diluted to a concentration predicted to produce near confluent lysis after infection of the appropriate host.”

Line 134: Change Discovery to Isolation

Done

--------4 comments below were addressed in first paragraph of reply-------------

Line 58: remove the sentence „We’ve engaged students in phage hunting using Bacillus species as host bacteria for several years, due to the ubiquitous nature of Bacillus in the environment as well as the ease of propagation on simple media.”

Line 84: there is not matter if students or researchers took samples, especially if they are co-authors or contributors to this paper.

Line 112: remove students

“in courses” was added to clarify the annotation work done through research-based SEA PHAGES courses.

Line 135 – 138: Rewrite the sentence with no information about students or Undergraduate researchers

The above four suggestions to remove mention of student involvement were not incorporated into the revisited manuscript and were addressed in the header paragraph to this reviewer.

Line 141: If all characterized phages are the same morphology please add a table with capsids and tails size to supplementary materials and additional information within the text that in Figure 1 are representative morphologies.

We have provided additional images for Figure 1. We do not have TEM images for all of these phage samples, and several images we do have are lower quality, so we are unfortunately unable to provide a table with capsid and tail sizes.

Line 493: Caudovirales should be italicized

Done

Line 516: Bacillus should be italicized

Done

Reviewer 3 Report

This manuscript is very nicely written and carefully conducted.There are two suggestions:

1.The TEM picture is blurred.Change a clear one.

2.Usually people are using VIRIDIC for calculating the intergenomic similarities

between phages(http://rhea.icbm.uni-oldenburg.de/VIRIDIC/).

Author Response

This manuscript is very nicely written and carefully conducted.There are two suggestions:

We thank the reviewer for careful reading and their suggestions to improve the manuscript.

1.The TEM picture is blurred.Change a clear one.

The figure has been updated with a more clear image for SerPounce as well as the addition of BeachBum.

2.Usually people are using VIRIDIC for calculating the intergenomic similarities

between phages(http://rhea.icbm.uni-oldenburg.de/VIRIDIC/).

This reviewer is correct that some researchers exploring virus taxonomy use VIRIDIC, as well as a variety of other tools to compare sequences. We compared our ANI values to the Salasmaviridae taxonomy proposal, and believe they provide similar outcomes. We have additionally incorporated a variety of other approaches that visualize and capture many of the smaller details between our genomes. We hope our approach is satisfactory and has provided novel information about these genomes that wouldn’t be captured through VIRIDIC.